# New Subsampling Algorithms for Fast Least Squares Regression

**Paramveer S. Dhillon**[1]   **Yichao Lu**[2]   **Dean Foster**[2]   **Lyle Ungar**[1]
[1]Computer & Information Science, [2]Statistics (Wharton School)
University of Pennsylvania, Philadelphia, PA, U.S.A
`{dhillon|ungar}@cis.upenn.edu`
`foster@wharton.upenn.edu, yichaolu@sas.upenn.edu`

## Abstract

We address the problem of fast estimation of ordinary least squares (OLS) from large amounts of data ($n \gg p$). We propose three methods which solve the big data problem by subsampling the covariance matrix using either a single or two stage estimation. All three run in the order of size of input i.e. O($np$) and our best method, *Uluru*, gives an error bound of $O(\sqrt{p/n})$ which is independent of the amount of subsampling as long as it is above a threshold. We provide theoretical bounds for our algorithms in the fixed design (with Randomized Hadamard preconditioning) as well as sub-Gaussian random design setting. We also compare the performance of our methods on synthetic and real-world datasets and show that if observations are i.i.d., sub-Gaussian then one can directly subsample without the expensive Randomized Hadamard preconditioning without loss of accuracy.

## 1   Introduction

Ordinary Least Squares (OLS) is one of the oldest and most widely studied statistical estimation methods with its origins tracing back over two centuries. It is the workhorse of fields as diverse as Machine Learning, Statistics, Econometrics, Computational Biology and Physics. To keep pace with the growing amounts of data ever faster ways of estimating OLS are sought. This paper focuses on the setting ($n \gg p$), where $n$ is the number of observations and $p$ is the number of covariates or features, a common one for web scale data.

Numerous approaches to this problem have been proposed [1, 2, 3, 4, 5]. The predominant approach to solving big data OLS estimation involves using some kind of random projections, for instance, transforming the data with a randomized Hadamard transform [6] or Fourier transform and then uniformly sampling observations from the resulting transformed matrix and estimating OLS on this smaller data set. The intuition behind this approach is that these frequency domain transformations uniformize the data and smear the signal across all the observations so that there are no longer any high leverage points whose omission could unduly influence the parameter estimates. Hence, a uniform sampling in this transformed space suffices. Another way of looking at this approach is as preconditioning the design matrix with a carefully constructed data-independent random matrix before subsampling. This approach has been used by a variety of papers proposing methods such as the Subsampled Randomized Hadamard Transform (SRHT) [1, 4] and the Subsampled Randomized Fourier Transform (SRFT) [2, 3]. There is also publicly available software which implements these ideas [7]. It is worth noting that these approaches assume a fixed design setting.

Following this line of work, in this paper we provide two main contributions:

1. **Novel Subsampling Algorithms for OLS:** We propose three novel[1] algorithms for fast estimation of OLS which work by subsampling the covariance matrix. Some recent results in [8] allow us to bound the difference between the parameter vector ($\widehat{w}$) we estimate from the subsampled data and the *true* underlying parameter ($w_0$) which generates the data. We provide theoretical analysis of our algorithms in the fixed design (with Randomized Hadamard preconditioning) as well as sub-Gaussian random design setting. The error bound of our best algorithm, *Uluru*, is independent of the fraction of data subsampled (above a minimum threshold of sub-sampling) and depends only on the characteristics of the data/design matrix $\mathbf{X}$.

2. **Randomized Hadamard preconditioning not always needed:** We show that the error bounds for all the three algorithms are similar for both the fixed design and the sub-Gaussian random design setting. In other words, one can either transform the data/design matrix via Randomized Hadamard transform (fixed design setting) and then use any of our three algorithms or, if the observations are i.i.d. and sub-Gaussian, one can directly use any of our three algorithms. Thus, another contribution of this paper is to show that if the observations are i.i.d. and sub-Gaussian then one does not need the slow Randomized Hadamard preconditioning step and one can get similar accuracies much faster.

The remainder of the paper is organized as follows: In the next section, we formally define notation for the regression problem, then in Sections 3 and 4, we describe our algorithms and provide theorems characterizing their performance. Finally, we compare the empirical performance of our methods on synthetic and real world data.

## 2 Notation and Preliminaries

Let $\mathbf{X}$ be the $n \times p$ design matrix. For the random design case we assume the rows of $\mathbf{X}$ are $n$ i.i.d samples from the $1 \times p$ independent variable (a.k.a. "covariates" or "predictors") $X$. $\mathbf{Y}$ is the real valued $n \times 1$ response vector which contains $n$ corresponding values of the dependent variable $Y$ (in general we use bold letter for samples and normal letter for random variables or vectors). $\epsilon$ is the $n \times 1$ homoskedastic noise vector with common variance $\sigma^2$. We want to infer $w_0$ i.e. the $p \times 1$ population parameter vector that generated the data.

More formally, we can write the true model as:
$$\mathbf{Y} = \mathbf{X}w_0 + \epsilon$$
$$\epsilon \sim_{iid} \mathcal{N}(0, \sigma^2)$$
The sample solution to the above equation (in matrix notation) is given by $\widehat{w}_{\text{sample}} = (\mathbf{X}^\top \mathbf{X})^{-1}\mathbf{X}^\top \mathbf{Y}$ and by consistency of the OLS estimator we know that $\widehat{w}_{\text{sample}} \rightarrow_d w_0$ as $n \rightarrow \infty$. Classical algorithms to estimate $\widehat{w}_{\text{sample}}$ use QR decomposition or bidiagonalization [9] and they require $\text{O}(np^2)$ floating point operations.

Since our algorithms are based on subsampling the covariance matrix, we need some extra notation. Let $r = n_{\text{subs}}/n \; (< 1)$ be the subsampling ratio, giving the ratio of the number of observations ($n_{\text{subs}}$) in the subsampled matrix $\mathbf{X}_{\text{subs}}$ fraction to the number of observations ($n$) in the original $\mathbf{X}$ matrix. I.e., $r$ is the fraction of the observations sampled. Let $\mathbf{X}_{\text{rem}}$, $\mathbf{Y}_{\text{rem}}$ denote the data and response vector for the remaining $n - n_{\text{subs}}$ observations. In other words $\mathbf{X}^\top = [\mathbf{X}_{\text{subs}}^\top \; ; \; \mathbf{X}_{\text{rem}}^\top]$ and $\mathbf{Y}^\top = [\mathbf{Y}_{\text{subs}}^\top \; ; \; \mathbf{Y}_{\text{rem}}^\top]$.

Also, let $\mathbf{\Sigma}_{XX}$ be the covariance of $X$ and $\mathbf{\Sigma}_{XY}$ be the covariance between $X$ and $Y$. Then, for the fixed design setting $\mathbf{\Sigma}_{XX} = \mathbf{X}^\top \mathbf{X}/n$ and $\mathbf{\Sigma}_{XY} = \mathbf{X}^\top \mathbf{Y}/n$ and for the random design setting $\mathbf{\Sigma}_{XX} = \mathbb{E}(\mathbf{X}^\top \mathbf{X})$ and $\mathbf{\Sigma}_{XY} = \mathbb{E}(\mathbf{X}^\top \mathbf{Y})$.

The bounds presented in this paper are expressed in terms of the Mean Squared Error (or Risk) for the $\ell_2$ loss. For the fixed design setting,
$$MSE = (w_0 - \widehat{w})^\top \mathbf{X}^\top \mathbf{X}(w_0 - \widehat{w})/n = (w_0 - \widehat{w})^\top \mathbf{\Sigma}_{XX}(w_0 - \widehat{w})$$
For the random design setting
$$MSE = \mathbb{E}_X \|Xw_0 - X\widehat{w}\|^2 = (w_0 - \widehat{w})^\top \mathbf{\Sigma}_{XX}(w_0 - \widehat{w})$$

## 2.1 Design Matrix and Preconditioning

Thus far, we have not made any assumptions about the design matrix $\mathbf{X}$. In fact, our algorithms and analysis work for both fixed design and random design settings.

As mentioned earlier, our algorithms involve subsampling the observations, so we have to ensure that we do not leave behind any observations which are outliers/high leverage points; This is done differently for fixed and random designs. For the fixed design setting the design matrix $\mathbf{X}$ is arbitrary and may contain high leverage points. Therefore before subsampling we precondition the matrix by a Randomized Hadamard/Fourier Transform [1, 4] and after conditioning, the probability of having high leverage points in the new design matrix becomes very small. On the other hand, if we assume $\mathbf{X}$ to be random design and its rows are i.i.d. draws from some nice distribution like sub-Gaussian, then the probability of having high leverage points is very small and we can happily subsample $\mathbf{X}$ without preconditioning.

In this paper we analyze both the fixed as well as sub-Gaussian random design settings. Since the fixed design analysis would involve transforming the design matrix with a preconditioner before subsampling, some background on SRHT is warranted.

**Subsampled Randomized Hadamard Transform (SRHT):** In the fixed design setting we precondition and subsample the data with a $n_{\text{subs}} \times n$ randomized hadamard transform matrix $\Theta (= \sqrt{\frac{n}{n_{\text{subs}}}} \mathbf{RHD})$ as $\Theta \cdot \mathbf{X}$.

The matrices $\mathbf{R}$, $\mathbf{H}$, and $\mathbf{D}$ are defined as:

- $\mathbf{R} \in \mathbb{R}^{n_{\text{subs}} \times n}$ is a set of $n_{subs}$ rows from the $n \times n$ identity matrix, where the rows are chosen uniformly at random without replacement.

- $\mathbf{D} \in \mathbb{R}^{n \times n}$ is a random diagonal matrix whose entries are independent random signs, i.e. random variables uniformly distributed on $\{\pm 1\}$.

- $\mathbf{H} \in \mathbb{R}^{n \times n}$ is a normalized Walsh-Hadamard matrix, defined as: $H_n = \begin{bmatrix} H_{n/2} & H_{n/2} \\ H_{n/2} & -H_{n/2} \end{bmatrix}$

  with, $H_2 = \begin{bmatrix} +1 & +1 \\ +1 & -1 \end{bmatrix}$. $H = \frac{1}{\sqrt{n}} H_n$ is a rescaled version of $H_n$.

It is worth noting that $\mathbf{HD}$ is the preconditioning matrix and $\mathbf{R}$ is the subsampling matrix.

The running time of SRHT is $n \ p \ log(p)$ floating point operations (FLOPS) [4]. [4] mention fixing $n_{subs} = O(p)$. However, in our experiments we vary the amount of subsampling, which is not something recommended by their theory. With varying subsampling, the run time becomes $O(n \ p \ log(n_{\text{subs}}))$.

## 3 Three subsampling algorithms for fast linear regression

All our algorithms subsample the $\mathbf{X}$ matrix followed by a single or two stage fitting and are described below. The algorithms given below are for the random design setting. The algorithms for the fixed design are exactly the same as below, except that $\mathbf{X}_{subs}$, $\mathbf{Y}_{subs}$ are replaced by $\Theta \cdot \mathbf{X}$, $\Theta \cdot \mathbf{Y}$ and $\mathbf{X}_{rem}$, $\mathbf{Y}_{rem}$ with $\Theta_{rem} \cdot \mathbf{X}$, $\Theta_{rem} \cdot \mathbf{Y}$, where $\Theta$ is the SRHT matrix defined in the previous section and $\Theta_{rem}$ is the same as $\Theta$, except that $\mathbf{R}$ is of size $n_{rem} \times n$. Still, for the sake of completeness, the algorithms are described in detail in the Supplementary material.

**Full Subsampling (FS):** Full subsampling provides a baseline for comparison; In it we simply r-subsample $(\mathbf{X}, \mathbf{Y})$ as $(\mathbf{X}_{\text{subs}}, \mathbf{Y}_{\text{subs}})$ and use the subsampled data to estimate both the $\mathbf{\Sigma}_{XX}$ and $\mathbf{\Sigma}_{XY}$ covariance matrices.

**Covariance Subsampling (CovS):** In Covariance Subsampling we r-subsample $\mathbf{X}$ as $\mathbf{X}_{\text{subs}}$ only to estimate the $\mathbf{\Sigma}_{XX}$ covariance matrix; we use all the $n$ observations to compute the $\mathbf{\Sigma}_{XY}$ covariance matrix.

**Uluru**  : *Uluru*[2] is a two stage fitting algorithm. In the first stage it uses the r-subsampled $(\mathbf{X}, \mathbf{Y})$ to get an initial estimate of $\widehat{w}$ (i.e., $\widehat{w}_{FS}$) via the Full Subsampling (FS) algorithm. In the second stage it uses the remaining data $(\mathbf{X}_{\text{rem}}, \mathbf{Y}_{\text{rem}})$ to estimate the bias of the first stage estimator $w_{correct} = w_0 - \widehat{w}_{FS}$. The final estimate ($w_{Uluru}$) is taken to be a weighted combination (generally just the sum) of the FS estimator and the second stage estimator ($\widehat{w}_{correct}$). *Uluru* is described in Algorithm 1.

In the second stage, since $\widehat{w}_{FS}$ is known, on the remaining data we have $\mathbf{Y}_{\text{rem}} = \mathbf{X}_{\text{rem}}w_0 + \epsilon_{\text{rem}}$, hence

$$
\begin{aligned}
\mathbf{R}_{\text{rem}} &= \mathbf{Y}_{\text{rem}} - \mathbf{X}_{\text{rem}} \cdot \widehat{w}_{FS} \\
&= \mathbf{X}_{\text{rem}}(w_0 - \widehat{w}_{FS}) + \epsilon_{\text{rem}}
\end{aligned}
$$

The above formula shows we can estimate $w_{correct} = w_0 - \widehat{w}_{FS}$ with another regression, i.e. $\widehat{w}_{correct} = (\mathbf{X}_{\text{rem}}^\top\mathbf{X}_{\text{rem}})^{-1}\mathbf{X}_{\text{rem}}^\top\mathbf{R}_{\text{rem}}$. Since computing $\mathbf{X}_{\text{rem}}^\top\mathbf{X}_{\text{rem}}$ takes too many FLOPS, we use $\mathbf{X}_{sub}^\top\mathbf{X}_{sub}$ instead (which has already been computed). Finally we correct $\widehat{w}_{FS}$ and $\widehat{w}_{correct}$ to get $\widehat{w}_{Uluru}$. The estimate $w_{correct}$ can be seen as an almost unbiased estimation of the error $w_0 - w_{\text{subs}}$, so we correct almost all the error, hence reducing the bias.

---

**Input**: $\mathbf{X}$, $\mathbf{Y}$, $r$
**Output**: $\widehat{w}$
$\widehat{w}_{FS} = (\mathbf{X}_{\text{subs}}^\top\mathbf{X}_{\text{subs}})^{-1}\mathbf{X}_{\text{subs}}^\top\mathbf{Y}_{\text{subs}}$;
$\mathbf{R}_{\text{rem}} = \mathbf{Y}_{\text{rem}} - \mathbf{X}_{\text{rem}} \cdot \widehat{w}_{FS}$;
$\widehat{w}_{correct} = \frac{n_{\text{subs}}}{n_{\text{rem}}} \cdot (\mathbf{X}_{\text{subs}}^\top\mathbf{X}_{\text{subs}})^{-1}\mathbf{X}_{\text{rem}}^\top\mathbf{R}_{\text{rem}}$;
$\widehat{w}_{Uluru} = \widehat{w}_{FS} + \widehat{w}_{correct}$;
**return** $\widehat{w} = \widehat{w}_{Uluru}$;

**Algorithm 1:** Uluru Algorithm

# 4 Theory

In this section we provide the theoretical guarantees of the three algorithms we discussed in the previous sections in the fixed as well as random design setting. All the theorems assume OLS setting as mentioned in Section 2. Without loss of generality we assume that $\mathbf{X}$ is whitened, i.e. $\mathbf{\Sigma}_{X,X} = \mathbf{I}_p$ (see Supplementary Material for justification). For both the cases we bound the square root of Mean Squared Error which becomes $\|w_0 - \widehat{w}\|$, as described in Section 2.

## 4.1 Fixed Design Setting

Here we assume preconditioning and subsampling with SRHT as described in previous sections. (**Please see the Supplementary Material for all the Proofs**)

**Theorem 1** *Assume $\mathbf{X} \in n \times p$ and $\mathbf{X}^\top\mathbf{X} = n \cdot \mathbf{I}_p$. Let $\mathbf{Y} = \mathbf{X}w_0 + \epsilon$ where $\epsilon \in n \times 1$ is i.i.d. gaussian noise with standard deviation $\sigma$.*

*If we use algorithm FS, then with failure probability at most $2\frac{n}{e^p} + 2\delta$,*

$$
\|w_0 - \hat{w}_{FS}\| \le C\sigma\sqrt{\ln(nr + 1/\delta)\frac{p}{nr}} \tag{1}
$$

**Theorem 2** *Assuming our data comes from the same model as Theorem 1 and we use CovS, then with failure probability at most $3\delta + 3\frac{n}{e^p}$,*

$$
\|w_0 - \hat{w}_{CovS}\| \le (1-r)\left(C_1\sqrt{\ln(\frac{2p}{\delta})\frac{p}{nr}} + C_2\sqrt{\ln(\frac{2p}{\delta})\frac{p}{n(1-r)}}\right)\|w_0\| + C_3\sigma\sqrt{\log(n+1/\delta)\frac{p}{n}} \tag{2}
$$

**Theorem 3** *Assuming our data comes from the same model as Theorem 1 and we use Uluru, then with failure probability at most $5\delta + 5\frac{n}{e^p}$,*

$$
\begin{aligned}
\|w_0 - \hat{w}_{Uluru}\| &\leq \sigma\sqrt{\ln(nr + 1/\delta)\frac{p}{nr}}\left(C_1\sqrt{\ln(\frac{2p}{\delta})\frac{p}{nr}} + C_2\sqrt{\ln(\frac{2p}{\delta})\frac{p}{n(1-r)}}\right) \\
&\quad + \sigma C_3\sqrt{\ln(n(1-r) + 1/\delta)\cdot\frac{p}{n(1-r)}}
\end{aligned}
$$

**Remark 1** *The probability $\frac{n}{e^p}$ becomes really small for large $p$, hence it can be ignored and the $\ln$ terms can be viewed as constants. Lets consider the case $n_{subs} \ll n_{rem}$, since only in this situation subsampling reduces computational cost significantly. Then, keeping only the dominating terms, the result of the above three theorems can be summarized as: With some failure probability less than some fixed number, the error of FS algorithm is $O(\sigma\sqrt{\frac{p}{nr}})$, the error of CovS algorithm is $O(\sqrt{\frac{p}{nr}}\|w\| + \sigma\sqrt{\frac{p}{n}})$ and the error of Uluru algorithm is $O(\sigma\frac{p}{nr} + \sigma\sqrt{\frac{p}{n}})$*

## 4.2 Sub-gaussian Random Design Setting

### 4.2.1 Definitions

The following two definitions from [10] characterize what it means to be sub-gaussian.

**Definition 1** *A random variable $X$ is sub-gaussian with sub-gaussian norm $\|X\|_{\psi_2}$ if and only if*

$$(E|X|^p)^{1/p} \leq \|X\|_{\psi_2}\sqrt{p} \qquad \text{for all } p \geq 1 \tag{3}$$

*Here $\|X\|_{\psi_2}$ is the minimal constant for which the above condition holds.*

**Definition 2** *A random vector $X \in R^n$ is sub-gaussian if the one dimensional marginals $x^\top X$ are sub-gaussian for all $x \in \mathbb{R}^n$. The sub-gaussian norm of random vector $X$ is defined as*

$$\|X\|_{\psi_2} = \sup_{\|x\|^2 = 1} \|x^\top X\|_{\psi_2} \tag{4}$$

**Remark 2** *Since the sum of two sub-gaussian variables is sub-gaussian, it is easy to conclude that a random vector $\mathbf{X} = (X_1, ..X_p)^\top$ is a sub-gaussian random vector when the components $X_1, ..X_p$ are sub-gaussian variables.*

### 4.2.2 Sub-gaussian Bounds

Under the assumption that the rows of the design matrix $\mathbf{X}$ are i.i.d draws for a $p$ dimensional sub-Gaussian random vector $X$ with $\Sigma_{XX} = I_p$, we have the following bounds (**Please see the Supplementary Material for all the Proofs**):

**Theorem 4** *If we use the FS algorithm, then with failure probability at most $\delta$,*

$$\|w_0 - \hat{w}_{FS}\| \leq C\sigma\sqrt{\frac{p\cdot\ln(2p/\delta)}{nr}} \tag{5}$$

**Theorem 5** *If we use the CovS algorithm, then with failure probability at most $\delta$,*

$$
\begin{aligned}
\|w_0 - \hat{w}_{CovS}\| &\leq (1-r)\left(C_1\sqrt{\frac{p}{n\cdot r}} + C_2\sqrt{\frac{p}{n(1-r)}}\right)\|w_0\| \\
&\quad + C_3\sigma\sqrt{\frac{p\cdot\ln(2(p+2)/\delta)}{n}}
\end{aligned}
\tag{6}
$$

**Theorem 6** *If we use Uluru, then with failure probability at most $\delta$,*

$$\|w_0 - \widehat{w}_{Uluru}\| \leq C_1\sigma\sqrt{\frac{p\cdot\ln(2(2p+2)/\delta)}{n\cdot r}}\left[C_2\sqrt{\frac{p}{n\cdot r}}+C_3\sqrt{\frac{p}{(1-r)\cdot n}}\right]$$

$$+ \quad C_4\sigma\sqrt{\frac{p\cdot\ln(2(2p+2)/\delta)}{(1-r)\cdot n}}$$

**Remark 3** *Here also, the* ln *terms can be viewed as constants. Consider the case $r \ll 1$, since this is the only case where subsampling reduces computational cost significantly. Keeping only dominating terms, the result of the above three theorems can be summarized as: With failure probability less than some fixed number, the error of the FS algorithm is $O(\sigma\sqrt{\frac{p}{rn}})$, the error of the CovS algorithm is $O(\sqrt{\frac{p}{rn}}\|w\|+\sigma\sqrt{\frac{p}{n}})$ and the error of the Uluru algorithm is $O(\sigma\frac{p}{rn}+\sigma\sqrt{\frac{p}{n}})$. These errors are exactly the same as in the fixed design case.*

### 4.3 Discussion

We can make a few salient observations from the error expressions for the algorithms presented in Remarks 1 & 3.

The second term for the error of the *Uluru* algorithm does not contain $r$ at all. If it is the dominating term, which is the case if

$$r > O(\sqrt{p/n}) \tag{7}$$

then the error of *Uluru* is approximately $O(\sigma\sqrt{\frac{p}{n}})$, which is completely independent of $r$. Thus, if $r$ is not too small (i.e., when Eq. 7 holds), the error bound for *Uluru* is not a function of $r$. In other words, when Eq. 7 holds, we do not increase the error by using less data in estimating the covariance matrix in *Uluru*. FS Algorithm does not have this property since its error is proportional to $\frac{1}{\sqrt{r}}$.

Similarly, for the CovS algorithm, when

$$r > O(\frac{\|w_0\|^2}{\sigma^2}) \tag{8}$$

the second term dominates and we can conclude that the error does not change with $r$. However, Eq. 8 depends on how large the standard deviation $\sigma$ of the noise is. We can assume $\|w_0\|^2 = O(p)$ since it is $p$ dimensional. Hence if $\sigma \leq O(\sqrt{p})$, Eq. 8 fails since it implies $r > O(1)$ and the error bound of CovS algorithm increases with $r$.

To sum this up, *Uluru* has the nice property that its error bound does not increase as $r$ gets smaller as long as $r$ is greater than a threshold. This threshold is completely independent of how noisy the data is and only depends on the characteristics of the design/data matrix $(n, p)$.

### 4.4 Run Time complexity

Table 1 summarizes the run time complexity and theoretically predicted error bounds for all the methods. We use these theoretical run times (FLOPS) in our plots.

## 5 Experiments

In this section we elucidate the relative merits of our methods by comparing their empirical performance on both synthetic and real world datasets.

### 5.1 Methodology

We can compare our algorithms by allowing them each to have about $O(np)$ CPU time (ignoring the log factors). This is of order the same time as it takes to read the data. Our target accuracy is $\sqrt{p/n}$, namely what a full least squares algorithm would generate. We will assume $n \gg p$. The

| Methods | Running Time | Error |
| | O(FLOPS) | bound |
| --- | --- | --- |
| **OLS** | $O(n\,p^2)$ | $O(\sqrt{p/n})$ |
| **FS** | $O(n_{\mathrm{subs}}\,p^2)$ | $O(\sqrt{p/n_{\mathrm{subs}}})$ |
| **CovS** | $O(n_{\mathrm{subs}}\,p^2 + n\,p)$ | * |
| **Uluru** | $O(n_{\mathrm{subs}}\,p^2 + n\,p)$ | $O(\sqrt{p/n})$ |
| **SRHT-FS** | $O(max(n\,p\,log(p), n_{\mathrm{subs}}\,p^2))$ | $O(\sqrt{p^2/n})$ |
| **SRHT-CovS** | $O(max(n\,p\,log(p), n_{\mathrm{subs}}\,p^2 + n\,p))$ | * |
| **SRHT-Uluru** | $O(max(n\,p\,log(p), n_{\mathrm{subs}}\,p^2 + n\,p))$ | $O(\sqrt{p/n})$ |

Table 1: Runtime complexity. $n_{\mathrm{subs}}$ is the number of observations in the subsample, $n$ is the number of observations, and $p$ is the number of predictors. * indicates that no uniform error bounds are known.

subsample size, $n_{\mathrm{subs}}$, for FS should be $O(n/p)$ to keep the CPU time $O(np)$, which leads to an accuracy of $\sqrt{p^2/n}$. For the CovS method, the accuracy depends on how noisy our data is (i.e. how big $\sigma$ is). When $\sigma$ is large, it performs as well as $\sqrt{p/n}$, which is the same as full least squares. When $\sigma$ is small, it performs as poorly as $\sqrt{p^2/n}$. For *Uluru*, to keep the CPU time $O(np)$, $n_{\mathrm{subs}}$ should be $O(n/p)$ or equivalently $r = O(1/p)$. As stated in the discussions after the theorems, when $r \geq O(\sqrt{p/n})$ (in this set up we want $r = O(1/p)$, which implies $n \geq O(p^3)$), *Uluru* has error bound $O(\sqrt{p/n})$ no matter what signal noise ratio the problem has.

## 5.2 Synthetic Datasets

We generated synthetic data by distributing the signal uniformly across all the $p$ singular values, picking the $p$ singular values to be $\lambda_i = 1/i^2, i = 1 : p$, and further varying the amount of signal.

## 5.3 Real World Datasets

We also compared the performance of the algorithms on two UCI datasets [3]: CPUSMALL (n=8192, p=12) and CADATA (n=20640, p=8) and the PERMA sentiment analysis dataset described in [11] (n=1505, p=30), which uses LR-MVL word embeddings [12] as features. [4]

## 5.4 Results

The results for synthetic data are shown in Figure 1 (top row) and for real world datasets are also shown in Figure 1 (bottom row).

To generate the plots, we vary the amount of data used in the subsampling, $n_{\mathrm{subs}}$, from $1.1p$ to $n$. For FS, this simply means using a fraction of the data; for CovS and *Uluru*, only the data for the covariance matrix is subsampled. We report the Mean Squared Error (MSE), which in the case of squared loss is the same as the risk, as was described in Section 2. For the real datasets we do not know the true population parameter, $w_0$, so we replace it with its consistent estimator $w_{MLE}$, which is computed using standard OLS on the entire dataset.

The horizontal gray line in the figures is the overfitting point; it is the error generated by $\hat{w}$ vector of all zeros. The vertical gray line is the $n \cdot p$ point; thus anything which is faster than that must look at only some of the data.

Looking at the results, we can see two trends for the synthetic data. Firstly, our algorithms with no preconditioning are much faster than their counterparts with preconditioning and give similar accuracies. Secondly, as we had expected, CovS performs best for high noise setting being slightly better than *Uluru*, and *Uluru* is significantly better for low noise setting.

For real world datasets also, *Uluru* is almost always better than the other algorithms, both with and without preconditioning. As earlier, the preconditioned alternatives are slower.

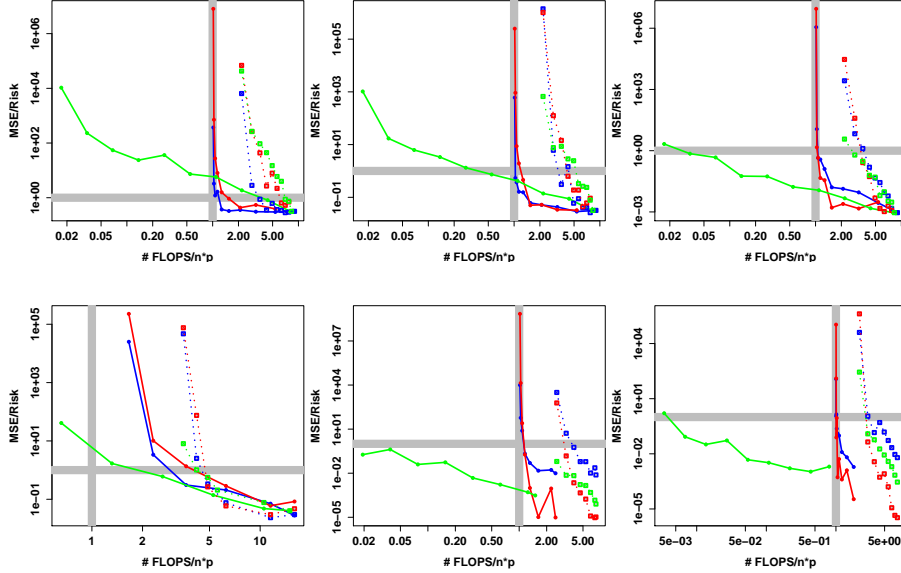

Figure 1: Results for synthetic datasets (n=4096, p=8) in top row and for (PERMA, CPUSMALL, CADATA, left-right) bottom row. The three columns in the top row have different amounts of signal, 2, $\sqrt{\frac{n}{p}}$ and $\frac{n}{p}$ respectively. In all settings, we varied the amount of subsampling from 1.1 $p$ to $n$ in multiples of 2. Color scheme: + (Green)-FS, + (Blue)-CovS, + (Red)-Uluru. The solid lines indicate no preconditioning (i.e. random design) and dashed lines indicate fixed design with Randomized Hadamard preconditioning. The FLOPS reported are the theoretical values (see Supp. material), the actual values were noisy due to varying load settings on CPUs.

# 6 Related Work

The work that comes closest to our work is the set of approaches which precondition the matrix by either Subsampled Randomized Hadamard Transform (SRHT) [1, 4], or Subsampled Randomized Fourier Transform (SRFT) [2, 3], before subsampling uniformly from the resulting transformed matrix.

However, this line of work is different our work in several ways. They are doing their analysis in a mathematical set up, i.e. solving an overdetermined linear system ($\hat{w} = \arg\min_{w \in \mathbb{R}^p} \|\mathbf{X}w - Y\|^2$), while we are working in a statistical set up (a regression problem $Y = \mathbf{X}\beta + \epsilon$) which leads to different error analysis.

Our FS algorithm is essentially the same as the subsampling algorithm proposed by [4]. However, our theoretical analysis of it is novel, and furtheremore they only consider it in fixed design setting with Hadamard preconditioning.

The CovS and *Uluru* are entirely new algorithms and as we have seen differ from FS in a key sense, namely that CovS and *Uluru* make use of all the data but FS uses only a small proportion of the data.

# 7 Conclusion

In this paper we proposed three subsampling methods for faster least squares regression. All three run in O(size of input) = $np$. Our best method, *Uluru*, gave an error bound which is independent of the amount of subsampling as long as it is above a threshold.

Furthermore, we argued that for problems arising from linear regression, the Randomized Hadamard transformation is often not needed. In linear regression, observations are generally i.i.d. If one further assumes that they are sub-Gaussian (perhaps as a result of a preprocessing step, or simply because they are 0/1 or Gaussian), then subsampling methods without a Randomized Hadamard transformation suffice. As shown in our experiments, dropping the Randomized Hadamard transformation significantly speeds up the algorithms and in i.i.d sub-Gaussian settings, does so without loss of accuracy.

## Footnotes

[1]One of our algorithms (FS) is similar to [4] as we describe in Related Work. However, even for that algorithm, our theoretical analysis is novel.

[2]*Uluru* is a rock that is shaped like a quadratic and is solid. So, if your estimate of the quadratic term is as solid as *Uluru*, you do not need use more data to make it more accurate.

[3] http://www.csie.ntu.edu.tw/~cjlin/libsvmtools/datasets/regression.html

[4] We also compared our approaches against coordinate ascent methods from [13] and our algorithms outperform them. Due to paucity of space we relegated that comparison to the supplementary material.

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
