[Supplementary Material]

# Supplementary Material for 'New Subsampling Algorithms for Fast Least Squares Regression'

**Paramveer S. Dhillon**[1]    **Yichao Lu**[2]    **Dean Foster**[2]    **Lyle Ungar**[1]

[1]Computer & Information Science, [2]Statistics (Wharton School)
University of Pennsylvania, Philadelphia, PA, U.S.A
{dhillon|ungar}@cis.upenn.edu
foster@wharton.upenn.edu, yichaolu@sas.upenn.edu

Firstly, we would like to state some lemmas and give some properties of Subsampled Randomized Hadamard Transform (SRHT), which will be pivotal in proving our theorems for the fixed design setting.

## 1 Properties of SRHT

As described in the paper, let $\mathbf{H}$ be the scaled Hadamard matrix of size $n \times n$, $\mathbf{D}$ be the diagonal matrix of size $n \times n$ with i.i.d. rademacher random variable on the diagonal and let $\mathbf{R} \in n_{subs} \times n$ be the subsampling matrix. So, $\Theta = \mathbf{RHD} \in n_{subs} \times n$ is the SRHT matrix. All the norms used in this paper and supplementary material are $\ell_2$ norms for a vector and the spectral norm for a matrix unless specified otherwise. The statement of the lemma is as follows:

**Lemma 1.** *Let $\mathbf{X}$ be an $n \times p$ $(n \gg p)$ matrix where $\mathbf{X}^\top \mathbf{X} = n \cdot \mathbf{I}_p$. Let $\Theta$ be a $n_{subs} \times n$ SRHT matrix where $n_{subs}$ is the subsampling size. Then with failure probability at most $\delta + \frac{n}{e^p}$,*

$$\|(\Theta\mathbf{X})^\top \Theta\mathbf{X}/n_{subs} - \mathbf{X}^\top\mathbf{X}/n\| \leq \sqrt{\frac{c\log(\frac{2p}{\delta})p}{n_{subs}}} \tag{1}$$

**Remark 1.** *The idea and tools for the proof of this lemma come from [1] and [2]. Here we characterize the spectral norm error between the matrix multiplication with and without SRHT as a function of subsample size $n_{subs}$ and matrix dimension $p$.*

Before proving Lemma 1 we need to state a few lemmas from random matrix theory. Next Lemma is Lemma 3.3 in [1].

**Lemma 2.** *(Row norms after Randomized Hadamard Transform) Let $\mathbf{V}$ be an $n \times p$ matrix with orthonormal columns. Then $\mathbf{HDV}$ is also an $n \times p$ matrix with orthonormal columns and*

$$\mathbf{P}\left(\max_{j=1,2...n} \|e_j^\top(\mathbf{HDV})\| \geq \sqrt{\frac{p}{n}} + \sqrt{\frac{8\log(\beta n)}{n}}\right) \leq \frac{1}{\beta} \tag{2}$$

**Remark 2.** *In our setting $p$ is reasonably large, though it's much smaller than $n$. Let $\beta = \frac{e^p}{n}$, we have $\max_{j=1,2...n} \|e_j^\top(\mathbf{HDV})\| \leq 4\sqrt{\frac{p}{n}}$ holds with failure probability at most $\frac{n}{e^p}$. In particular, when $\log(n) \ll p$ the failure probability is almost 0.*

Next lemma is Lemma 3.4 in [1] the proof of which comes from the matrix Chernoff bound in [2].

**Lemma 3.** *(Spectral Bounds for Row Sampling). Let $\mathbf{W}$ be an $n \times p$ matrix with orthonormal columns. Define $\mathbf{M} = n \cdot \max_{j=1,2...n} \|e_j^T W\|^2$. Draw $n_{subs}$ rows from $\mathbf{W}$ without replacement. Let $\mathbf{R} \in n_{subs} \times n$ be the matrix corresponding to subsampled rows. Then the smallest and largest*

*spectral value of the subsampled matrix* $\mathbf{RW}$ *are bounded by*

$$\sqrt{\frac{(1-\delta)n_{subs}}{n}} \quad \leq \quad \sigma_p(\mathbf{RW}) \tag{3}$$

$$\sqrt{\frac{(1+\eta)n_{subs}}{n}} \quad \geq \quad \sigma_1(\mathbf{RW}) \tag{4}$$

*with failure probability at most*

$$p \cdot \left( \frac{e^{-\delta}}{(1-\delta)^{1-\delta}} \right)^{n_{subs}/\mathbf{M}} + p \cdot \left( \frac{e^{\eta}}{(1+\eta)^{1+\eta}} \right)^{n_{subs}/\mathbf{M}} \tag{5}$$

Lemma 3 can be simplified a lot for our purpose.

**Corollary 1.** *Let* $\mathbf{W}$ *be an* $n \times p$ *matrix with orthonormal columns. Define* $\mathbf{M} = n \cdot \max_{j=1,2...n} \|e_j^\top \mathbf{W}\|^2$. *Draw* $n_{subs}$ *rows from* $\mathbf{W}$ *without replacement. Let* $\mathbf{R} \in n_{subs} \times n$ *be the matrix corresponding to the subsampled rows. Then the spectral values of the subsampled matrix* $\mathbf{RW}$ *are bounded by*

$$\sqrt{\frac{(1-\delta)n_{subs}}{n}} \quad \leq \quad \sigma_p(\mathbf{RW}) \tag{6}$$

$$\sqrt{\frac{(1+\delta)n_{subs}}{n}} \quad \geq \quad \sigma_1(\mathbf{RW}) \tag{7}$$

*with failure probability at most*

$$2p \cdot e^{\frac{-c\delta^2 n_{subs}}{\mathbf{M}}} \tag{8}$$

*for some fixed positive constant c.*

*Proof.* By the Taylor's expansion of $\log(1-\delta)$ and $\log(1+\delta)$

$$\log \left( \frac{e^{-\delta}}{(1-\delta)^{1-\delta}} \right) \quad = \quad -\delta - (1-\delta)\log(1-\delta) \leq -\delta^2$$

$$\log \left( \frac{e^{\delta}}{(1+\delta)^{1+\delta}} \right) \quad = \quad \delta - (1+\delta)\log(1+\delta) \leq -\delta^2/4$$

replace the $\frac{e^{-\delta}}{(1-\delta)^{1-\delta}}$ and $\frac{e^{\eta}}{(1+\eta)^{1+\eta}}$ term in lemma 2 with $e^{-c\delta^2}$ and $e^{-c\eta^2}$. Set $\eta = \delta$ completes the proof. $\square$

Now we can prove Lemma 1:

*Proof.* $\Theta = \mathbf{RHD}$. Let $\mathbf{W} = \mathbf{HDX}$, note that the columns of $\mathbf{X}/\sqrt{n}$ are orthonormal. Remark 2 shows

$$\max_{j=1,2...n} \|e_j^\top \mathbf{W}/\sqrt{n}\| \leq 4\sqrt{\frac{p}{n}} \tag{9}$$

holds with failure probability $\frac{n}{e^p}$. Let $\mathbf{M} = 16p = n \cdot \max_{j=1,2...n} \|e_j^\top \mathbf{W}/\sqrt{n}\|^2$. Assume equation 9 holds, Corollary 1 implies the spectral norm of $\Theta \mathbf{X}/\sqrt{n} = \mathbf{RW}/\sqrt{n}$ can be bounded by

$$\sqrt{\frac{(1-\varepsilon)n_{subs}}{n}} \quad \leq \quad \sigma_p(\Theta \mathbf{X}/\sqrt{n}) \tag{10}$$

$$\sqrt{\frac{(1+\varepsilon)n_{subs}}{n}} \quad \geq \quad \sigma_1(\Theta \mathbf{X}/\sqrt{n}) \tag{11}$$

with failure probability at most $\delta$ where $\varepsilon = \sqrt{\frac{c \log(\frac{2p}{\delta})p}{n_{subs}}}$. Equations 10, 11 implies that the singular values of the symmetric matrix $\frac{(\Theta \mathbf{X})^\top \Theta \mathbf{X}}{n}$ lie between $[\frac{(1-\varepsilon)n_{subs}}{n}, \frac{(1+\varepsilon)n_{subs}}{n}]$, or in other words,

the singular values of the symmetric matrix $\frac{(\Theta\mathbf{X})^\top\Theta\mathbf{X}}{n_{subs}}$ lies between $[1-\varepsilon, 1+\varepsilon]$. Noticing that $\mathbf{X}^\top\mathbf{X}/n$ is a $p \times p$ identity matrix, so Equations 10, 11 directly imply Equation 1. Finally let's compute the failure probability, i.e. the probability that the Equations 10, 11 don't hold. By Lemma 1,

$$P(\text{Equation 9 fails}) \leq \frac{n}{e^p} \tag{12}$$

By corollary 1,

$$P(\text{One of Equations 10, 11 fail}|\text{Equation 9 holds}) \leq \delta \tag{13}$$

which directly implies

$$P(\text{One of Equations 10, 11 fail and Equation 9 holds}) \leq \delta \tag{14}$$

Equations 12, 14 imply

$$
\begin{aligned}
P(\text{One of Equations 10, 11 fail}) \quad &\leq \quad P(\text{One of Equations 10, 11 fail and Equation 9 holds}) \\
&\quad + P(\text{Equation 9 fails }) \\
&\leq \quad \frac{n}{e^p} + \delta
\end{aligned}
$$

$\square$

Next two lemmas gives bounds on $(\Theta\mathbf{X})^\top\Theta\epsilon$ for SRHT, where $\epsilon$ is a $n \times 1$ i.i.d. centered gaussian noise with standard deviation $\sigma$.

**Lemma 4.** *Consider a finite sequence of fixed matrices $\{\mathbf{B}_k\}$ with dimension $d_1 \times d_2$, and let $\{\gamma_k\}$ be a finite sequence of independent standard normal variables. Define parameter*

$$c_1^2 = \max\{\|\sum_k \mathbf{B}_k\mathbf{B}_k^\top\|, \|\sum_k \mathbf{B}_k^\top\mathbf{B}_k\|\} \tag{15}$$

*Then for all $t > 0$*

$$P\{\|\sum_k \gamma_k\mathbf{B}_k\| \geq t\} \leq (d_1 + d_2)\cdot e^{-t^2/2c_1^2} \tag{16}$$

Lemma 4 comes from [2]. Now, Lemma 2 and 4 directly imply

**Lemma 5.** *With failure probability at most $\delta + \frac{n}{e^p}$,*

$$(\Theta\mathbf{X})^\top\Theta\epsilon/n_{subs} \leq \sigma\sqrt{log(\frac{n_{subs}+1}{\delta})\cdot 32\frac{p}{n_{subs}}} \tag{17}$$

*Proof.* Let $\mathbf{B}_k$ be the $k^{th}$ column of $(\Theta\mathbf{X})^\top$. Lemma 2 and Remark 2 imply $\max_k \|\mathbf{B}_k/\sqrt{n}\| \leq 4\sqrt{\frac{p}{n}}$ with failure probability at most $\frac{n}{e^p}$. In this case,

$$\|\sum_k \mathbf{B}_k\mathbf{B}_k^\top\| \leq \sum_k \|\mathbf{B}_k\mathbf{B}_k^\top\| \leq \sum_k \|\mathbf{B}_k\|\|\mathbf{B}_k^\top\| \leq 16pn_{subs}$$

Same bound holds for $\|\sum_k \mathbf{B}_k^\top\mathbf{B}_k\|$. On the other hand, $\Theta\epsilon$ is a $n_{subs} \times 1$ vector the elements of which are $n_{subs}$ i.i.d centered normal random variables with variance $\sigma$. Applying Lemma 4 with $c_1^2 = 16pn_{subs}$, we can bound the failure probability and using the same technique as Lemma 1 proves this Lemma. $\square$

We also need a lemma from matrix perturbation theory for the accuracy of the matrix inverse, which is given in Theorem 2.5 in [3]

**Lemma 6.** *Let $\Sigma_X$ be a non-singular $p \times p$ matrix and $\widehat{\Sigma}_X = \Sigma_X + \Delta_1$. Let $\|.\|$ be the matrix 2 norm. If $\widehat{\Sigma}_X$ is non-singular, then*

$$\frac{\|\widehat{\Sigma}_X^{-1} - \Sigma_X^{-1}\|}{\|\widehat{\Sigma}_X^{-1}\|} \leq \|\Sigma_X^{-1}\Delta_1\| \tag{18}$$

**Remark 3.** *In our model, $\Sigma_X = \mathbf{X}^\top \mathbf{X}/n = \mathbf{I}_p$. When $\|\Delta_1\| \leq \frac{1}{2}$, $\|\widehat{\Sigma}_X^{-1}\| \leq 2$. So we have, when $\Sigma_X = \mathbf{I}_p$ and $\|\Delta_1\|$ are small enough,*

$$\|\widehat{\Sigma}_X^{-1} - \Sigma_X^{-1}\| \leq C\|\Delta_1\| \tag{19}$$

*for some constant $C$.*

Lemmas 1 and 6 directly imply:

**Corollary 2.** *Let $(\mathbf{X}\Theta^\top \Theta \mathbf{X}/n_{subs})^{-1}$ be the estimator of the inverse of covariance, $\mathbf{I}_p = (\mathbf{X}^\top \mathbf{X}/n)^{-1}$. With failure probability at most $\frac{n}{e^p} + \delta$*

$$\|(\mathbf{X}\Theta^\top \mathbf{X}\Theta/l)^{-1} - \mathbf{I}_p\| \leq C\sqrt{\log(\frac{2p}{\delta})\frac{p}{n_{subs}}}$$

*for some constant $C$ if $n_{subs}$ is big enough.*

## 2 Proof for The Fixed Design Setting

For fixed design setting, let $\Theta = \mathbf{R}\mathbf{H}\mathbf{D} \in n_{subs} \times n$ be the SRHT where $\mathbf{R}$ is the row sampling matrix, $\mathbf{H}$ is the normalized Hadamard matrix and $\mathbf{D}$ is the diagonal random sign matrix. Let $r = \frac{n_{subs}}{n}$ denote the subsampling ratio. Define $\Theta_{rem} = \mathbf{R}_{rem}\mathbf{H}\mathbf{D}$ where $\mathbf{R}_{rem}$ samples those $n_{rem} = n - n_{subs}$ rows that are not included by $\mathbf{R}$. For notational convenience let $\Theta_{all} = (\Theta; \Theta_{rem})$ which can be viewed as a $n \times n$ SRHT matrix. Our three sampling estimators are then:

---
**Algorithm 1** Full Subsampling (FS)

    1. $\hat{w}_{FS} = ((\Theta \mathbf{X})^\top \Theta \mathbf{X})^{-1} (\Theta \mathbf{X})^\top \Theta Y$

---

---
**Algorithm 2** Covariance Subsampling (CovS)

    1. $\hat{w}_{CovS} = \frac{n_{subs}}{n} ((\Theta \mathbf{X})^\top \Theta \mathbf{X})^{-1} \mathbf{X}^\top Y$

---

---
**Algorithm 3** Uluru

    1. $\hat{w}_{FS} = ((\Theta \mathbf{X})^\top \Theta \mathbf{X})^{-1} (\Theta \mathbf{X})^\top \Theta Y$
    2. $R_{rem} = \Theta_{rem} Y - \Theta_{rem} \mathbf{X}\hat{w}_{FS}$
    3. $\hat{w}_{correct} = \frac{n_{subs}}{n_{rem}} \cdot ((\Theta \mathbf{X})^\top \Theta \mathbf{X})^{-1} (\Theta_{rem}\mathbf{X})^\top R_{rem}$,
    4. $\hat{w}_{Uluru} = \hat{w}_{FS} + \hat{w}_{correct}$ is our final estimator.

---

Now let's prove bounds for the fixed design setting:

### 2.1 Proof of Theorem 1

*Proof.*

$$\begin{aligned}
\|\hat{w}_{FS} - w\| &= \|(\mathbf{X}\Theta^\top \Theta \mathbf{X})^{-1}\mathbf{X}\Theta^\top \Theta^\top Y - w\| \\
&= \|(\mathbf{X}\Theta^\top \Theta \mathbf{X})^{-1}\mathbf{X}\Theta^\top \Theta \mathbf{X}w - w + (\mathbf{X}\Theta^\top \Theta \mathbf{X})^{-1}\mathbf{X}\Theta^\top \Theta \epsilon\| \\
&\leq \|(\mathbf{X}\Theta^\top \Theta \mathbf{X})^{-1}\|\|\mathbf{X}\Theta^\top \Theta \epsilon\|
\end{aligned}$$

With failure probability $\frac{n}{e^p} + \delta$ the inequality in Corollary 2 holds, $\|(\mathbf{X}\Theta^\top \Theta X/n_{subs})^{-1}\|$ is bounded by some constant. On the other hand, by Lemma 5 $\|\mathbf{X}\Theta^\top \Theta \epsilon/n_{subs}\|$ is bounded by $\sigma\sqrt{\log(nr + 1/\delta) \cdot 32\frac{p}{nr}}$ with failure probability $\frac{n}{e^p} + \delta$. Applying union bounds finishes the proof. $\qquad\square$

## 2.2 Proof for Theorem 2

*Proof.* Let $\Delta_2 = \mathbf{I} - (\mathbf{X}^\top \Theta^\top \Theta \mathbf{X}/n_{subs})^{-1}$ and $\Delta_3 = \mathbf{I} - (\mathbf{X}^\top \Theta_{rem}^\top \Theta_{rem} \mathbf{X}/(n_{rem}))$

$$
\begin{aligned}
\|w - \hat{w}_{CovS}\| &= \|\frac{n_{subs}}{n}(\mathbf{X}^\top \Theta^\top \Theta \mathbf{X})^{-1}\mathbf{X}^\top Y - w\| \\
&= \|\frac{n_{subs}}{n}((\mathbf{X}^\top \Theta^\top \Theta \mathbf{X})^{-1}(\mathbf{X}^\top \Theta^\top, \mathbf{X}^\top \Theta_{rem}^\top)((\Theta \mathbf{X}w; \Theta_{rem}\mathbf{X}w) + \Theta_{all}\epsilon) - w\| \\
&= \|\frac{n_{subs}}{n}(\mathbf{X}^\top \Theta^\top \Theta \mathbf{X})^{-1}(\mathbf{X}^\top \Theta^\top \Theta \mathbf{X}w) + \frac{n_{subs}}{n}(\mathbf{X}^\top \Theta^\top \Theta \mathbf{X})^{-1}(\mathbf{X}^\top \Theta_{rem}^\top \Theta_{rem}\mathbf{X}w) \\
&\quad + \frac{n_{subs}}{n}(\mathbf{X}^\top \Theta^\top \Theta \mathbf{X})^{-1}\mathbf{X}^\top \Theta_{all}^\top \Theta_{all}\epsilon - w\| \\
&\leq (1-r)\|(\mathbf{X}^\top \Theta^\top \Theta \mathbf{X}/n_{subs})^{-1}(\mathbf{X}^\top \Theta_{rem}^\top \Theta_{rem}\mathbf{X}w/n_{rem}) - w\| \\
&\quad + \|(\mathbf{X}^\top \Theta^\top \Theta \mathbf{X}/n_{subs})^{-1}\mathbf{X}\Theta_{all}^\top \Theta_{all}\epsilon/n\| \\
&\leq (1-r)\|(\mathbf{I} - \Delta_2)(\mathbf{I} - \Delta_3) - I\|\|w\| + \|(\mathbf{X}^\top \Theta^\top \Theta \mathbf{X}/n_{subs})^{-1}\mathbf{X}^\top \Theta_{all}^\top \Theta_{all}\epsilon/n\| \\
&\leq (1-r)(\|\Delta_2\| + \|\Delta_3\| + \|\Delta_2\Delta_3\|)\|w\| + \|(\mathbf{X}^\top \Theta^\top \Theta \mathbf{X}/n_{subs})^{-1}\|\|\mathbf{X}^\top \Theta_{all}^\top \Theta_{all}\epsilon/n\|
\end{aligned}
$$

By Corollary 2, $\Delta_2 \leq C\sqrt{\log(\frac{2p}{\delta})\frac{p}{nr}}$ with failure probability $\frac{n}{e^p} + \delta$. By Lemma 1, $\Delta_3 \leq C\sqrt{\log(\frac{2p}{\delta})\frac{p}{n(1-r)}}$ with failure probability $\frac{n}{e^p} + \delta$. By Lemma 5, $\|\mathbf{X}^\top \Theta_{all}^\top \Theta_{all}\epsilon/n\| \leq \sigma\sqrt{\log(n + 1/\delta).32\frac{p}{n}}$ with failure probability $\frac{n}{e^p} + \delta$. Also we can conclude that $\|(\mathbf{X}^\top \Theta^\top \Theta \mathbf{X}/n_{subs})^{-1}\|$ is bounded by some constant when the bound in corollary 2 holds. Ignore the $\|\Delta_2\Delta_3\|$ term since it's ignorable compared with other terms. Using the union bound we can prove Theorem 2. □

## 2.3 Proof for Theorem 3

*Proof.*

$$
\begin{aligned}
\|\hat{w}_{Uluru} - w\| &= \|(\mathbf{X}^\top \Theta^\top \Theta \mathbf{X})^{-1}\mathbf{X}^\top \Theta^\top \Theta Y + \hat{w}_{correct} - w\| \\
&= \|\hat{w}_{FS} + \left(\frac{r}{1-r}(\mathbf{X}^\top \Theta^\top \Theta \mathbf{X})^{-1}X^\top \Theta_{rem}^\top (\Theta_{rem}\mathbf{X}w + \Theta_{rem}\epsilon - \Theta_{rem}\mathbf{X}\hat{w}_{FS})\right) - w\| \\
&\leq \|\hat{w}_{FS} + \left(\frac{r}{1-r}(\mathbf{X}^\top \Theta^\top \Theta \mathbf{X})^{-1}(\mathbf{X}^\top \Theta_{rem}^\top \Theta_{rem}\mathbf{X})(w - \hat{w}_{FS})\right) - w\| \\
&\quad + \|\frac{r}{1-r}(\mathbf{X}^\top \Theta^\top \Theta \mathbf{X})^{-1}\mathbf{X}^\top \Theta_{rem}^\top \Theta_{rem}\epsilon\| \\
&= \|\hat{w}_{FS} - w + ((\mathbf{X}^\top \Theta^\top \Theta \mathbf{X}/n_{subs})^{-1}(\mathbf{X}^\top \Theta_{rem}^\top \Theta_{rem}\mathbf{X}/n_{rem})(w - \hat{w}_{FS}))\| \\
&\quad + \|(\mathbf{X}^\top \Theta^\top \Theta \mathbf{X}/n_{subs})^{-1}\mathbf{X}^\top \Theta_{rem}^\top \Theta_{rem}\epsilon/n_{rem}\| \\
&= \|\hat{w}_{FS} - w\|\|\mathbf{I} - (\mathbf{I} - \Delta_2)(\mathbf{I} - \Delta_3)\| + \|(\mathbf{X}^\top \Theta^\top \Theta \mathbf{X}/n_{subs})^{-1}\mathbf{X}^\top \Theta_{rem}^\top \Theta_{rem}\epsilon/n_{rem}\| \\
&\leq \|\hat{w}_{FS} - w\|(\|\Delta_2\| + \|\Delta_3\| + \|\Delta_2\Delta_3\|) \\
&\quad + \|(\mathbf{X}^\top \Theta^\top \Theta \mathbf{X})^{-1}\|\|\mathbf{X}^\top \Theta_{rem}^\top \Theta_{rem}\epsilon/n_{rem}\|
\end{aligned}
$$

By Theorem 1, $\|\hat{w}_{FS} - w\| \leq C\sigma\sqrt{\ln(nr + 1/\delta)32\frac{p}{nr}}$ with failure probability $2\frac{n}{e^p} + 2\delta$. By Corollary 2, $\Delta_2 \leq C\sqrt{\log(\frac{2p}{\delta})\frac{p}{nr}}$ with failure probability $\frac{n}{e^p} + \delta$. By Lemma 3, $\Delta_3 \leq C\sqrt{\log(\frac{2p}{\delta})\frac{p}{n(1-r)}}$ with failure probability $\frac{n}{e^p} + \delta$. By lemma 5, $\|\mathbf{X}^\top \Theta_{rem}^\top \Theta_{rem}\epsilon/(n(1-r))\| \leq \sigma\sqrt{\log(n(1-r) + 1/\delta) \cdot 32\frac{p}{n(1-r)}}$ with failure probability $\frac{n}{e^p} + \delta$. Also we can conclude that $\|(\mathbf{X}^\top \Theta^\top \Theta X/n_{subs})^{-1}\|$ is bounded by some constant when the bound in Corollary 2 holds. Applying union bound proves Theorem 3. □

# 3 Subgaussian Random Design Setting

In this section we proof bounds for three algorithm in the subgaussian random design setting.

The first lemma is a straightforward generalization of Equation 5.25 in [4] and provides error bounds for the empirical estimator of $\Sigma_{XX}$.

**Lemma 7.** *Let $\widehat{\Sigma}_{\mathrm{subs}} \equiv (\mathbf{X}_{\mathrm{subs}}^\top \mathbf{X}_{\mathrm{subs}})/n_{\mathrm{subs}}$ be the estimator of covariance matrix $\Sigma_{XX}$ with sample size $n_{\mathrm{subs}}$. Then, with probability at least $1 - \delta$,*

$$\|\widehat{\Sigma}_{\mathrm{subs}} - \Sigma_{XX}\| \le C_1 \sqrt{\frac{p}{n_{\mathrm{subs}}}} + C_2 \frac{\sqrt{\ln(2/\delta)}}{\sqrt{n_{\mathrm{subs}}}}$$

*if $n_{\mathrm{subs}} \gg \ln(2/\delta)$ and $n_{\mathrm{subs}} \gg p$.*
*Assume $\ln(2/\delta) < p$, the above bound can be simplified to*

$$\|(\mathbf{X}_{\mathrm{subs}}^\top \mathbf{X}_{\mathrm{subs}})/n_{\mathrm{subs}} - \Sigma_{XX}\| \le C \sqrt{\frac{p}{n_{\mathrm{subs}}}}. \tag{20}$$

The next lemma gives a concentration bound on the centered sub-exponential variables, which is a slightly modified version of corollary 5.17 in [4].

**Lemma 8.** *Let $z_1 ... z_n$ be i.i.d. draws from centered sub-exponential random variables. For $\epsilon \le K$,*

$$P\left(\frac{|\sum_{i=1}^n z_i|}{n} > \epsilon\right) \le 2 \exp\left\{-\frac{c\epsilon^2 n}{K^2}\right\} \tag{21}$$

*In other words, with probability at least $1 - \delta$,*

$$\frac{|\sum_{i=1}^n z_i|}{n} \le K \sqrt{\frac{\ln(2/\delta)}{cn}} \tag{22}$$

*if $n \ge \frac{\ln(2/\delta)}{c}$.*

Here $c$ is an absolute constant and $K$ is the sub-exponential norm of the random variable $z_i$. If the variables $z_i$ is scaled by a constant $\lambda$, then the sub-exponential norm is also scaled by $\lambda$.

Lemma 8 directly implies the following:

**Corollary 3.** *Let $\mathbf{X}_{\mathrm{subs}} \in n_{\mathrm{subs}} \times p$ be $n_{\mathrm{subs}}$ i.i.d. draws from a $p$ dimensional subgaussian random vector and let $\epsilon_{\mathrm{subs}} \in n_{\mathrm{subs}} \times 1$ be i.i.d. draws from a centered normal with standard deviation $\sigma$. Then, for big enough $n_{\mathrm{subs}}$, with probability $1 - \delta$,*

$$\|\mathbf{X}_{\mathrm{subs}}^\top \epsilon_{\mathrm{subs}}/n_{\mathrm{subs}}\| \le C\sigma \sqrt{\frac{p \cdot \ln(2p/\delta)}{n_{\mathrm{subs}}}} \tag{23}$$

*Proof.* Every element in the $p$ dimensional vector $\mathbf{X}_{\mathrm{subs}}^\top \epsilon_{\mathrm{subs}}/n_{\mathrm{subs}}$ can be viewed as the mean of $n_{subs}$ i.i.d. centered sub-exponential variable with sub-exponential norm proportional to $\sigma$. Now, by applying Lemma to every element of $\mathbf{X}_{\mathrm{subs}}^\top \epsilon_{\mathrm{subs}}/n_{\mathrm{subs}}$ and using the union bound, the corollary follows. $\square$

Similar as corollary 2 for the fixed design setting, Lemmas 6 and 7 directly imply:

**Corollary 4.** *Let $\widehat{\Sigma}_{\mathrm{subs}}^{-1} = (\mathbf{X}_{\mathrm{subs}}^\top \mathbf{X}_{\mathrm{subs}}/n_{\mathrm{subs}})^{-1}$ be the estimator of the inverse of covariance of $\mathbf{X}$, i.e. $\Sigma_{XX}^{-1}$. Then, with probability $1 - \delta$*

$$\|\widehat{\Sigma}_{\mathrm{subs}}^{-1} - \Sigma_{XX}^{-1}\| \le C \sqrt{\frac{p}{n_{\mathrm{subs}}}}$$

*for some constant $C$ if $n_{\mathrm{subs}}$ is big enough.*

With lemmas and theorems listed above, we can prove the three theorems stated in the main text.

### 3.1 Proof of Theorem 4

$$
\begin{aligned}
\|\hat{w}_{FS} - w_0\| &= \|(\mathbf{X}_{\text{subs}}^\top \mathbf{X}_{\text{subs}})^{-1}\mathbf{X}_{\text{subs}}^\top Y_{\text{subs}} - w_0\| \\
&= \|(\mathbf{X}_{\text{subs}}^\top \mathbf{X}_{\text{subs}})^{-1}\mathbf{X}_{\text{subs}}^\top \mathbf{X}_{\text{subs}} w_0 - w_0 + (\mathbf{X}_{\text{subs}}^\top \mathbf{X}_{\text{subs}})^{-1}\mathbf{X}_{\text{subs}}^\top \epsilon_{\text{subs}}\| \\
&\leq \|(\mathbf{X}_{\text{subs}}^\top \mathbf{X}_{\text{subs}}/n_{\text{subs}})^{-1}\| \cdot \|\mathbf{X}_{\text{subs}}^\top \epsilon_{\text{subs}}/n_{\text{subs}}\|
\end{aligned}
$$

Since $\|(\mathbf{X}_{\text{subs}}^\top \mathbf{X}_{\text{subs}}/n_{\text{subs}})^{-1}\|$ can be bounded by some constant, applying Corollary 3, and noting that $n_{\text{subs}} = nr$, we can prove Theorem 4.

### 3.2 Proof of Theorem 5

Define $\widehat{\Sigma}_{\text{rem}} \equiv \mathbf{X}_{\text{rem}}^\top \mathbf{X}_{\text{rem}}/n_{\text{rem}}$. First note that $\hat{w}_{CovS} - w_0$ is[1]

$$
= (1-r)(\widehat{\Sigma}_{\text{subs}}^{-1}\widehat{\Sigma}_{\text{rem}} - I)w_0 + r\widehat{\Sigma}_{\text{subs}}^{-1}\mathbf{X}^\top \epsilon/n
$$

Let $\widehat{\Sigma}_{\text{subs}}^{-1} = I + \Delta_4$ and $\widehat{\Sigma}_{\text{rem}} = I + \Delta_5$, So we have $\hat{w}_{CovS} - w_0$ is

$$
(1-r)(\Delta_4 + \Delta_5 + \Delta_4\Delta_5)w_0 + r\widehat{\Sigma}_{\text{subs}}^{-1}\mathbf{X}^\top \epsilon/n
$$

We can bound $\|\hat{w}_{CovS} - w_0\|$ by

$$
\begin{aligned}
&\leq (1-r)(\|\Delta_4\| + \|\Delta_5\| + \|\Delta_4\Delta_5\|)\|w_0\| \\
&\quad + r\|\widehat{\Sigma}_{\text{subs}}^{-1}\|\|\mathbf{X}^\top \epsilon/n\|
\end{aligned}
$$

Bounding $\Delta_4$ with Corollary 4, bounding $\Delta_5$ with Lemma 7 and bounding $\|\mathbf{X}^\top \epsilon/n\|$ with corollary 3, and using the union bound we can prove Theorem 5.

### 3.3 Proof of Theorem 6

$$
\|\hat{w}_{Uluru} - w_0\| = \|(\mathbf{X}_{\text{subs}}^\top \mathbf{X}_{\text{subs}})^{-1}\mathbf{X}_{\text{subs}}^\top Y_{\text{subs}} + \theta\hat{w}_{correct} - w_0\|
$$

$$
\begin{aligned}
&= \left\|\hat{w}_{FS} + \theta\frac{r}{1-r}(\mathbf{X}_{\text{subs}}^\top \mathbf{X}_{\text{subs}})^{-1}\mathbf{X}_{\text{rem}}^\top(\mathbf{X}_{\text{rem}}w_0 + \epsilon_3 - \mathbf{X}_{\text{rem}}\hat{w}_{FS}) - w_0\right\| \\
&\leq \left\|\hat{w}_{FS} + \theta\frac{r}{1-r}(\mathbf{X}_{\text{subs}}^\top \mathbf{X}_{\text{subs}})^{-1}\mathbf{X}_{\text{rem}}^\top\mathbf{X}_{\text{rem}}(w_0 - \hat{w}_{FS}) - w_0\right\| + \theta\|(\mathbf{X}_{\text{subs}}^\top \mathbf{X}_{\text{subs}}/n_{\text{subs}})^{-1}\mathbf{X}_{\text{rem}}^\top\epsilon_{\text{rem}}/n_{\text{rem}}\| \\
&= \|\hat{w}_{FS} - w_0 + \theta(\mathbf{X}_{\text{subs}}^\top \mathbf{X}_{\text{subs}}/n_{\text{subs}})^{-1}\mathbf{X}_{\text{rem}}^\top\mathbf{X}_{\text{rem}}/n_{\text{rem}}(w_0 - \hat{w}_{FS})\| + \theta\|(\mathbf{X}_{\text{subs}}^\top \mathbf{X}_{\text{subs}}/n_{\text{subs}})^{-1}\mathbf{X}_{\text{rem}}^\top\epsilon_{\text{rem}}/n_{\text{rem}}\| \\
&= \|\hat{w}_{FS} - w_0\|\|I - \theta(I - \Delta_4)(I - \Delta_5)\| + \theta\|(\mathbf{X}_{\text{subs}}^\top \mathbf{X}_{\text{subs}}/n_{\text{subs}})^{-1}\mathbf{X}_{\text{rem}}^\top\epsilon_{\text{rem}}/n_{\text{rem}}\| \\
&\leq \|\hat{w}_{FS} - w_0\|(\|I - \theta I\| + \theta\|\Delta_4\| + \theta\|\Delta_5\|) + \theta\|\Delta_4\Delta_5\|) + \theta\|(\mathbf{X}_{\text{subs}}^\top \mathbf{X}_{\text{subs}}/n_{\text{subs}})^{-1}\|\|\mathbf{X}_{\text{rem}}^\top\epsilon_{\text{rem}}/n_{\text{rem}}\|
\end{aligned}
$$

Bounding $\|\hat{w}_{FS} - w_0\|$ with Theorem 1, bounding $\Delta_4$ with Corollary 4, bounding $\Delta_5$ with Lemma 7, bounding $\|\mathbf{X}_{\text{rem}}^\top\epsilon_{\text{rem}}/n_{\text{rem}}\|$ with Corollary 3, and using union bound we can prove Theorem 6.

## 4 Coordinate Free vs. Coordinate Based Approaches

Our algorithms presented in this paper are coordinate free. Another family of algorithms that are popular are coordinate based.

$$
\begin{aligned}
w_0 - \hat{w}_{CovS} &= r(\mathbf{X}_{\text{subs}}^\top \mathbf{X}_{\text{subs}})^{-1}\mathbf{X}^\top \mathbf{Y} - w_0 \\
&= r(\mathbf{X}_{\text{subs}}^\top \mathbf{X}_{\text{subs}})^{-1}(\mathbf{X}_{\text{subs}}^\top, \mathbf{X}_{\text{rem}}^\top)((\mathbf{X}_{\text{subs}}w_0; \mathbf{X}_{\text{rem}}w_0) + \epsilon) - w_0 \\
&= r(\mathbf{X}_{\text{subs}}^\top \mathbf{X}_{\text{subs}})^{-1}(\mathbf{X}_{\text{subs}}^\top \mathbf{X}_{\text{subs}}w_0) + r(\mathbf{X}_{\text{subs}}^\top \mathbf{X}_{\text{subs}})^{-1}(\mathbf{X}_{\text{rem}}^\top\mathbf{X}_{\text{rem}}w_0) + r(\mathbf{X}_{\text{subs}}^\top \mathbf{X}_{\text{subs}})^{-1}\mathbf{X}^\top \epsilon - w_0 \\
&= r(\mathbf{X}_{\text{subs}}^\top \mathbf{X}_{\text{subs}})^{-1}(\mathbf{X}_{\text{rem}}^\top\mathbf{X}_{\text{rem}}w_0) + r(\mathbf{X}_{\text{subs}}^\top \mathbf{X}_{\text{subs}})^{-1}\mathbf{X}^\top \epsilon - (1-r)w_0 \\
&= (1-r)\widehat{\Sigma}_{\text{subs}}^{-1}\widehat{\Sigma}_{\text{rem}}w_0 - (1-r)w_0 + r\widehat{\Sigma}_{\text{subs}}^{-1}\mathbf{X}^\top \epsilon/n \\
&= (1-r)(\widehat{\Sigma}_{\text{subs}}^{-1}\widehat{\Sigma}_{\text{rem}} - I)w_0 + r\widehat{\Sigma}_{\text{subs}}^{-1}\mathbf{X}^\top \epsilon/n
\end{aligned}
$$

**Coordinate Based Approaches:**   Recently,  [5] have shown that stochastic dual coordinate ascent (SDCA) methods enjoy stronger theoretical guarantees for large scale machine learning compared to stochastic gradient descent (SGD). They consider the regularized loss minimization problem of the form:

$$P(w) = \frac{1}{n} \left[ \sum_{i=1}^{n} \phi_i(w^\top x_i) + \frac{\lambda}{2} \|w\|^2 \right].$$

In the experiments we report below, we use the squared loss, i.e., $\phi_i(a) = (a - y_i)^2$. Their method works in the dual space and makes a few passes over the data (a few "epochs"). In each epoch, they either randomly pick an instance (called SDCA in their paper) or permute the set of instances (called SDCA-PERM). They then perform a simple additive update to the dual parameters followed by averaging. The number of epochs (N) and the regularization constant ($\lambda$) are tunable constants. In addition to the better convergence bounds, another attractive feature of their approach is that there is a definitive stopping criteria based on duality gap ($\epsilon_D$): i.e., one should stop when the duality gap reaches a certain pre-chosen value. The run time of SDCA based approaches can be seen to be $O(n\,p\,N)$, where $N$ is the number of epochs.

A main difference between their gradient based approach and our subsampling approach is that their approach depends on the underlying coordinates of the system. Below we compare against both their random and permutation based SDCA variants, which we call (SDCA-r) and (SDCA-p) respectively.

## 5   Extra Experiments

Here we present some additional experiments benchmarking the performance of our algorithms against stochastic dual coordinate ascent (SDCA) methods.

We generated synthetic data from two models, further varying the amount of signal carried by each model.

- **Low Rank Data**: Here, we put the signal in the direction of the major principle components.

- **Coordinate-based Data**: This setting is designed to make coordinate based systems perform poorly. The features are divided in pairs where each feature is very highly negatively correlated with its pair, but the sum of the two features predicts $Y$ well. I.e. each pair of features is $x_i = \epsilon_i + \gamma\delta_j$, $x_{i+1} = \epsilon_i - \gamma\delta_j$ with the $\delta$'s being predictive of $Y$ and the $\epsilon$'s being much larger–but independent of $Y$. Thus, if you look at a simple regression (regression with only one feature), neither feature in the pair will look promising, but if you look at both, they are highly significant. An alternative way to think of this model is to that it has $p/2$ large singular values which carry no signal and $p/2$ small singular values whose directions carry all the signal.

The two settings were constructed such that SDCA would perform very well on one of them (Low Rank) and would perform poorly on the other one (Coordinate Based).

For the results presented below, we used $p = 8$ and $n = 4,096$ and tried different values for the signal present in the datasets $[2, \sqrt{\frac{n}{p}}, \frac{n}{p}]$. For SDCA, we tried $\lambda \in 0.01, 0.1$ and $1$, and chose the value which gave minimum MSE/Risk.

### 5.1   Results

As can be seen from the plots, SDCA-p performs better than our algorithms in the Low-Rank setting, especially when there was low signal. Otherwise our algorithms both in fixed as well as subgaussian random design setting performed better than SDCA-p. SDCA-r gave a similar performance.

Figure 1: Results for synthetic datasets (n=4096, p=8) in Low-Rank (top row) and Coordinate Based Data (bottom row). The three columns in the top row have different amounts of signal, 2, $\sqrt{\frac{n}{p}}$ and $\frac{n}{p}$ respectively. In all settings, we varied the amount of subsampling from $1.1\,p$ to $n$ in multiples of 2. Color scheme: + (Green)-FS, + (Blue)-CovS, + (Red)-Uluru ▪ (Black)-SDCA-p. The solid lines indicate no preconditioning (i.e. random design) and dashed lines indicate fixed design with Randomized Hadamard preconditioning.

## 6    Error Measures and Rotations

The bounds presented in this paper are expressed in terms of the Mean Squared Error (or Risk) for the $\ell_2$ loss; i.e., $E_{\mathbf{X}}\|\mathbf{X}^\top w_0 - \mathbf{X}^\top \widehat{w}\|^2$. Note that the expectation is w.r.t. unseen new observations $\mathbf{X}$ and not the training samples.

Now,

$$
\begin{aligned}
\mathbb{E}_{\mathbf{X}}\|\mathbf{X}^\top w_0 - \mathbf{X}^\top \widehat{w}\|^2 &= (w_0 - \widehat{w})^\top \mathbb{E}_{\mathbf{X}}(\mathbf{X}^\top \mathbf{X})(w_0 - \widehat{w}) \\
&= (w_0 - \widehat{w})^\top \mathbf{\Sigma}_{XX}(w_0 - \widehat{w})
\end{aligned}
\tag{24}
$$

Therefore, when $\mathbf{\Sigma}_{XX}$ is the identity matrix, $\mathbb{E}_{\mathbf{X}}\|\mathbf{X}^\top w_0 - \mathbf{X}^\top \widehat{w}\|^2 = \|w_0 - \widehat{w}\|^2$.

So, for mathematical convenience, when $\mathbf{\Sigma}_{XX}$ is an arbitrary positive semi-definite matrix, we can rotate the $\mathbf{X}$ matrix and $w_0$ to make $\mathbf{\Sigma}_{XX}$ identity.

Therefore, let:

$$
\begin{aligned}
w_0^{\text{new}} &= \mathbf{\Sigma}_{XX}^{1/2} w_0 \\
\mathbf{X}^{\text{new}} &= \mathbf{\Sigma}_{XX}^{-1/2}\mathbf{X} \\
Y &= \mathbf{X}^{\text{new}} w_0^{\text{new}} + \epsilon
\end{aligned}
$$

Since, all our three estimators are coordinate free, they have the property that transforming the estimator by $\widehat{w}^{\text{new}} = \mathbf{\Sigma}_{XX}^{1/2}\widehat{w}$ will generate the exact same estimate as computing the estimator on the transformed data. So, if we use the new transformed $\mathbf{X}$, our error estimates are identical. Hence, our error measure, Eq. 24, becomes $\|w_0^{\text{new}} - \widehat{w}^{\text{new}}\|^2$.

In the new transformed model, $\mathbf{X}^{\text{new}}$ has identity covariance. The above calculation indicates that in order to bound the error measure (Eq. 24) for $\mathbf{X}$ with arbitrary covariance structure, one can equivalently bound $\|w_0 - \widehat{w}\|^2$ when $\mathbf{\Sigma}_{XX} = \mathbf{I}$.

## Footnotes

[1]Follows by