[Reviews · NeurIPS 2013]

Submitted by Assigned_Reviewer_4

The paper studies algorithms for least squares regression when the error for each example is a random Gaussian based on fast Hadamard/Fourier transform. Previous analyses of similar algorithms focused on the worst case error setting rather than random. The paper shows that the new algorithms based on fast Hadamard transform with running time O(np log p), where n is the number of examples and p is the number of parameters, get the same expected error bound sqrt(p/n) as the standard (slower) algorithm. The proof is based on the proofs from previous work in the worst case setting.

The proofs are clearly written and seem correct. Since the focus of the paper is the proofs, the authors might want to at least include a sketch of the proofs in the main body. The results are probably useful in practice, where the error might be random rather than worst case. The experiments need some more work however: the settings of some experiments such as n=4096 and p=8 are outside of the effective range of the theory because n/e^p > 1 and there is no bound on the failure probability in this case. In all experiments involving real data (not synthetic), the baseline seems to outperform the new algorithms, thus raising doubt about the usefulness of the new algorithms. It would be useful if the authors present some experiment on real data where the new algorithms at least beat the baseline.

===========

Rebuttal comments: I give the paper the benefit of the doubt and believe that the new algorithms are better than the baseline on some data.
Summary: The paper studies performance of algorithms based on fast Hadamard transform for least square regression when the error is random Gaussian. The results might be useful in practice when the error might be random rather than worst case.

Submitted by Assigned_Reviewer_5

The paper compares three very simple algorithms for approximating OLS using less than O(np^2) computation, "full subsampling", "covariance subsampling" and "uluru". Each method uses the same input covariance matrix estimate based on an r-subsample, but handles the input-output co-variance slightly differently. Both fixed and random designs are considered. The approximate algorithms can be made to run in O(np), ie p times faster the OLS. The paper concludes with an experimental section with simulated and realistic data.

The main thrust of the paper is the analysis, showing how the error depends on various quantities, and I think is is a pity, or misjudgement not to include the analysis in the paper (three theorems a stated in the paper, but all analysis is only in the supplementary material).

The motivation is a bit unclear. What types of data are these methods suitable for? All the data sets shown are so small, that exact OLS can be performed without difficulty. If there indeed are problems of practical interest out there with sufficiently large n and p, that exact OLS cannot be done conveniently, then why don't you present a few of these? Also, it is not quite clear to me that the data sets presented really do lie solidly on the n and p ranges needed by the theory. The main result for uluru requires O(p^3) < n << exp(p), is this really fulfilled in the data presented?

The plots in figure 1 (main experimental results) are poor. The logarithmic y-axis spans from 1e-29 to > 1e6. Most of this scale extending over 35 orders of magnitude is not of practical interest; chosing such a large scale compresses the relevant region into such a small space that it becomes illegible.

It is
Summary: Three simple approximate methods for OLS are presented. It is not clear that the proposed algorithm is of significant practical value to the NIPS community, no theory is included in the paper. Only toy examples are given where OLS could be run exactly.

Submitted by Assigned_Reviewer_6

This paper proposes new subsampling algorithms for large scale least squares problems. They propose two new subsampling based algorithms and provide error bounds for these. They also provide error bound for a previously proposed algorithm. All the error bounds are in terms of estimation errors in the weight vector which is different from previous works.

The paper is very well written and the algorithms/results are novel. The results are specifically insightful in large data setting (n > > p) and say that if the subsampling factor is higher than a fixed threshold, the error is essentially independent of the subsampling factor.
Summary: A very well written paper with clear, novel contributions.
Author Feedback

Author rebuttal: We would like to thank all the reviewers for the insightful reviews.

Unfortunately, we couldn't rebut the incomplete sentence "It is.." by Assigned_Reviewer_5; and hope that there was no subsequent part of his/her review that was missing!


Assigned_Reviewer_4:

We would add a sketch of the proofs in the main body of the paper.

Regarding your comment about synthetic data, we would like to point out that n,p with n/e^p >1 was chosen intentionally as theory already dictates behavior in n/e^p < 1 case, however we thought it would be interesting to show the performance in the case when the assumptions of the theorem don't hold. As can be seen even in the n/e^p >1 case, the algorithms perform as expected. We agree that in order to be totally rigorous we should have a failure probability, as you rightly pointed out; we were trying to make that same point via simulation!

Also, we would replace the real-world results in the paper.

**
We added some real-world results in the final paper that underscore the points made by theory.
**

Assigned_Reviewer_5:

>>>The main thrust of the paper is the analysis, showing how the error depends on various quantities, and I think is is a pity, or misjudgement not to include the analysis in the paper (three theorems a stated in the paper, but all analysis is only in the supplementary material).
>>>
>>>
It is not clear that the proposed algorithm is of significant practical value to the NIPS community, no theory is included in the paper.
>>>


We had way too much stuff to fit into 8 pages, and we thought that theorem proofs should be relegated to the supplementary material, as is done by most other NIPS papers. As we mentioned in our reply to "Assigned_Reviewer_4", we would be happy to add a sketch of the proofs in the main body of the paper.


>>>
The motivation is a bit unclear. What types of data are these methods suitable for? All the data sets shown are so small, that exact OLS can be performed without difficulty.
>>>

There is a rich literature on efficient algorithms for large scale regression and this is an important problem. Please see the citations [1,2,3,4,5] and the references therein, in the paper which we also describe in Section 1; they are replete with real world examples of such settings as well as efficient algorithms for this setting.

The O(p^3) < n< e^p condition is easily satisfied in:

1). Text/NLP applications: where "n" is the number of words in a corpus typically ~ 1 billion and p ~ 50, where each dimension of "p" is a real valued embedding for that word, for instance, learned via Deep Learning (see the Deep Learning work on learning word embeddings coming out of Stanford, U. Toronto and U. Montreal) or Spectral Learning (see the work on spectral learning work on learning word representations coming out of U. Penn).

2). Marketing Applications: n ~ 100 million (e.g. the customers who have purchased from Amazon) and p~ 50, (information about their demographics and past purchasing behavior.)

Please note that there is no way that we could have put all the theorem proofs as well as perform extensive large scale experiments as you ask and fit all that in 8 pages.